# To Save a Girl-Child, You Must Train a Boy-Child: A Note on Situational Irony

**DOI:** 10.3390/ijerph192316313

**Published:** 2022-12-06

**Authors:** Emmanuel O. Amoo, Mercy E. Adebayo, Michael O. Owoeye, Matthew E. Egharevba

**Affiliations:** 1Demography and Social Statistics, Covenant University, Ota 112104, Nigeria; 2Department of Sociology, Covenant University, Ota 112104, Nigeria; 3Department of Sociology, Bowen University, Iwo 232101, Nigeria

**Keywords:** boy-child, girl-child, unwanted pregnancy, family planning, initiatives, girls/women sexual and reproductive health rights

## Abstract

Despite numerous initiatives and resources to save and protect the health and sexual rights of girls and women, the persistently high rate of unwanted pregnancy, abortion, and sexual violence in sub-Saharan Africa (SSA) has remain a topical public health challenge. This study hypothesised that the continuous conspicuous omission of boys/men in the interventions to combat this menace could be a long-life impediment to the realisation of sustainable health for girls and women in the region. The study adopted a systematic review of extant population-based published studies from Scopus, Google Scholars, PubMed, EMBASE, and AJOL. Literature coverage included the post-United Nations’ coordinated International Conference on Population and Development (ICPD), Cairo, 1994, which marked the beginning of a massive campaign for women/girls sexual rights. The obtained qualitative data were appraised and synthesised towards spurring policy recommendations for gender balanced initiatives on the sexual and reproductive health rights in SSA. The study highlighted that unwanted pregnancy occurs only when a boy/man has unprotected sex with a girl/woman without considering her choice or rights. It is considered ironic that the dominant factors are boys and men but many enlightenment initiatives/campaigns are concentrated on girls and women. The study developed a schematic save-a-girl-child framework that illustrated the possible dividends inherent in the training of a boy-child to achieve a safer world for the girls/women. It recommends increase in the exposure of boys and men to sexual education and counselling, which can motivate them to be supporters of family planning, supporters of only wanted pregnancy, wanted fatherhood, marital fidelity, intimate partners’ harmonious living rather than violence, and wife or partner empowerment.

## 1. Background

Irony is non-exclusively meaningful in theatrical parlance but widely used in day-to-day communication. Literarily, irony points to a situation where there is contrast between expectation and reality [1]. An irony could be likened to when there is a difference between what is being practiced and what is supposed to be the practice. In its extended form, irony could be dramatic, comedic, subverted, and underplaying the severity of a phenomenon [1]. In its adaptive form, initiative ironies are conjectured as the treatment of effects rather than the known underlying factors. Initiative irony could be described as a situation where prevention is less important than curing, and curing is less significant than palliative exercises. Most initiatives on girls’/women’s sexual reproductive issues are shrouded with ironies, where the source(s) of a challenge is/are neglected and the consequences are perennially discussed and treated. This study expresses concern over the ironies surrounding most initiatives that are tailored toward girls’/women’s sexual and reproductive rights. It juxtaposes a few of these initiatives with the current practices (that are seemingly neglected or permissive culturally) which could stand as barriers to the achievement of girls/women sexual and reproductive rights and good health and wellbeing for girls and women.

Since the massive campaign for girls’/women’s rights began decades ago and has received increased mainstream attention in public health care from the early 2000s [2,3], several initiatives to protect the female gender have continued to emerge, and a number of popular world fora have integrated campaigns for girls’/women’s rights into the mainstream attention of public health care [2,3]. We have had the World Conference on Human Rights in Vienna (14–25 June 1993) that signaled to the whole world that “women rights are human rights” [4], the United Nations’ coordinated International Conference on Population and Development (ICPD), Cairo in 1994, and the 1995 Beijing Conference that highlighted concerns on women and girls’ empowerment and autonomy [5]. There was also the Declaration on the Elimination of Violence against Women [6,7], with the annual “Orange the World” slogan from 2014 till date [8,9,10]. Notable among them is also the Convention on the elimination of all forms of discrimination against women (CEDAW) adopted by the United Nations General Assembly that encouraged the modification of all social and cultural conducts that could have negative impacts on the health and wellbeing of women [6,11,12]. There are also the 2000 Millennium Development Goals (MDGs), the current pursuits that are enshrined in the Sustainable Development Goals (SDGs 2015), and the African Union Agenda 2063. Great highlights in these initiatives are the elimination of girl-child marriage, forced marriage, and support for women empowerment, including free basic education for girls [5,13,14,15]. The initiatives also included laws and regulations against women trafficking and application of criminal laws to under-age sex, child-brides, and sex work [16,17]. 

### Highlights of Organisations Supporting Girls and Women Reproductive and Sexual Rights

In addition to the world conferences or for a on girl and women reproductive and sexual rights, there are also numerous organisations fighting for gender equality but with a bias towards women and girls’ interests. A review of history revealed over 20 international organisations in this respect. There is Plan International, which came into being in 1937, and has been working to advance the rights of children and equality for girls, and abhors discrimination against girls. We also have the International Alliance of Women (IAW) saddled with the responsibility of promoting human rights of women and girls globally [18,19]. Notable among these organisations is also the Gender at Work, a feminist knowledge network that works to build inclusive cultures and end discrimination against women.

The Global Fund for Women (founded in 1987) is among the world’s leading organizations that promotes, among other things, gender equality and human rights of girls/women, especially their sexual and reproductive health and rights. There is also the Association for Women’s Rights in Development (AWID). The association envisions a world where feminist realities flourish and there is equitable resources’ allocation. This also include the Womankind Worldwide (founded in 1989), which pursues the protection of women’s rights and the transformation of women’s lives. There is also the Center for Reproductive Rights that came into being in 1992 and has been using the power of the law to advance reproductive rights globally by also pursuing gender equality in access to quality reproductive healthcare, including birth control, safe abortion, prenatal and obstetric, and women’s autonomy in decision-making [18]. While the European Women’s Lobby (EWL) was founded in 1990 with a vision of a society in which the contribution of women to all aspects of life is recognized and celebrated, the global Women’s Environment and Development Organization (WEDO) founded in 1990 has remained an advocacy organization pursuing the protection of gender equality, human rights, and the integrity of the environment, in order to ensure that women’s voices are heard, and women’s leadership is advanced [18].

There is also Equality Now, which was founded in 1992 with the mission of using legal advocacy to protect and promote the human rights of women and girls. This is pivoted on recognition that sex discriminatory laws could be a fundamental impediment to the achievement of gender equality. In 1993, the Women for Women International was created with the conspicuous banner of ‘Bride and Not Girls’. The organisation supports marginalised women in countries affected by war and conflict. Likewise, the Rise Up group has been advancing women’s rights, equality, education, sexual and reproductive health, and economic empowerment since 2009 [18]. 

The list also includes the United Nations Entity for Gender Equality and the Empowerment of Women (known as UN Women) founded in 2010. The group is strongly dedicated to accelerating progress on meeting women’s needs across the world. The UN Women is an advocacy on policies, laws, and services that benefit women, especially in the areas of participation and taking leading roles in governance systems, decent work and economic autonomy, freedom from all forms of violence, among others [18,20]. The MATCH International Women’s Fund also gained more popularity in 2013 (thought, it has been established since 1976). This group is devoted to matching the needs of women with resources, especially through investment in businesses. Similarly, there is the Time’s Up group (founded in 2018) that is championing the creation of safe, fair, and dignified work for women of all kinds [18]. 

While the list of organisations pursuing women interests is inexhaustible, advocacies for men are scanty in the literature. However, despite these initiatives and organisation movements, the rates of unwanted pregnancy, abortion, and maternal deaths have not been conspicuously reduced in sub-Saharan African region [9,12,21,22]. While these innumerable measures were designed to save and protect girls and women, it is noted that these programmes and initiatives are explicitly concerned tangentially about the boys and men who are the suspected perpetrators. For every unwanted pregnancy, child-bride, abortion, etc., a boy/man is involved. Till date, there have not been popular behavioural suggestions for men concerning their own reproductive health rights; however, condom use has been entrenched [12,23,24] and including family planning [25,26]. This paper is an exposition of the uniqueness of balanced gender initiatives considering the basic truth that, to save a girl-child, you must train a boy-child.

## 2. Methods and Materials

### 2.1. Search Strategy and Selection Criteria

The study adopted a systematic review of extant population-based published studies that relate to reproductive and sexual health rights and wellbeing from Scopus, Google Scholars, PubMed, EMBASE, and AJOL. The strategy followed the Preferred Reporting Items for Systematic Reviews and Meta-Analyses (PRISMA) procedures [27,28,29]. The relevant qualitative data gathered were later appraised and the results synthesised towards spurring policy recommendations for gender balance initiatives on the reproductive health and sexual rights in sub-Saharan Africa.

### 2.2. Inclusion and Exclusion Criteria 

For inclusion and exclusion criteria, cross sectional studies, position papers, documents from relevant organisations, and governments from 2000 were included. The selected literature reviewed were mostly post-1994 International Conference on Population and Development (ICPD) publications where the world (179 governments) launched the historical sexual and reproductive health as a universal human rights. The post-ICPD marked the era of a massive campaign for women/girls’ reproductive health rights. However, where the complete article could not be downloaded, the titles and abstracts of such articles were reviewed for relevance and further screening [27]. The exclusion criteria were as follows: (1) studies or documents that are not human being based; (2) studies that are not related to reproductive and sexual health or rights are excluded; (3) studies without information on sub-Saharan Africa; and (4) non-relevant articles, reports, and so on. Data from the studies that met the requirement were extracted and assessed for quality. The entire retrieval procedures are as shown in Figure 1.

## 3. Findings and Discussion

The findings highlighted that the stakeholders often emphasise preventing adolescent pregnancy without recourse to the culprit for correctional counselling, punishment, or any other form of sanction. In the same society, fathering a child is applauded irrespective of the age of the father [30]. There are also the practice of multiple sexual partnerships or multiple spouses that are being treated as a demonstration of wealth in sub-Saharan Africa [31,32]. While a compulsory dropout decision (‘unwritten eviction law’) is meted out on pregnant girls from schools, the boy involved continues his education and, in most cases, without any form of reprimand from the society. Where a few countries within the region have laws preventing young girls from being married off [27], there are no known policies outlawing boy-child fatherhood. To date, there is no known popular law that frowns against an adolescent boy (or young man) inheriting a widow, even if such widow is much older than his mother. In the same society, where there is campaign against female genital cutting, it is a situational irony that virginity testing, virginal cleansing, and forced marriages are performed as cultural rites and, in some instances, these are done with open ceremonies [33,34,35].

Achieving a world where every pregnancy would be wanted has been the mission and clamour of numerous stakeholders in women and girls’ rights, health, and development with an inestimable investment or resources expended. Despite all these efforts, there is oversight on the fact that unwanted pregnancy may not exist where the men and boys have not engaged in sexual relationships with women or girls. Unwanted pregnancy may occur when the man or boy does not consider the choice or rights of the girl/woman concerned but successfully had the sex. It is therefore initiative irony when the dominant factor is the men, but much effort have been centering on girls/women at the expense of the boy/man that also requires adequate information on the unwanted pregnancy and its consequences. For instance, the initiatives on empowerment of women and girls’ rights to ‘say No’ (to rape or harassment, for example) could stop the presence but may not prevent the reoccurrence or further attempt by the same man and this time around with subtler arrangement(s). The seemingly effective initiative could be such training that could stop the present and educate the boys and men on why such acts should not be attempted in the future, a kind of enlightenment to make the boy or man recognising that it is a violation of a woman’s rights. 

Notwithstanding the pursuit of massive campaigns for family planning and contraceptives, there is flagrantly acquiescence to multiple wives’ practices within the same communities and cultures [36,37]. In this same society, the want of a child is traditionally obligatory either for an economic reason or cultural pride. In sub-Saharan Africa, numerous religions and cultures supported many wives while only a few mandated one wife for certain religious offices. The practice of having many wives is seemingly a permissiveness for wife-exchange or wife-replacement whenever the need arises. For example, in Nigeria, it is customarily expected that men of title should have several wives [27,38], except among the Christians, especially in the Pentecostals, where men that hold offices are not encouraged to have more than one wife. Some other religion affiliations (e.g., traditional and Islam) permit multiple wives [39]. However, it is ironic that, despite the fact that Christianity favours ‘one-man-one-wife’, a report has indicated that the risk of untimed pregnancy among women that practice traditional religion is relatively 56% less compared to the Christians and that the likelihood of experiencing unwanted pregnancy among Muslims is relatively 29% lower compared to the rates among the Christian women [40].

In addition, it is an irony that, as the region is fighting against girls/women sexual exploitation, the much newly celebrated notion on polyandry in South Africa permits a woman to have more than one husband at a time [37]. So, who is exploiting who? 

The condom distribution programmes could be termed as an appropriate initiative. The profound objectives and effectiveness lie in stopping or reducing unprotected sexual activity and its consequences such as unintended pregnancy and curbing the spread of sexually transmitted diseases (STDs) including HIV and AIDS [41,42]. However, the initiative, per se, is silenced on the limitation of the frequency of condom use. There seems to be an element of irony in the school-based condom disbursement initiative where innocent school girls and boys could be indirectly exposed to sexual materials. In addition, considering the curiosity of adolescents and young boys, holding or having access to a free condom could spur the attempt to use it. Notwithstanding this, having good knowledge of condom use is not synonymous with its utilisation and practice of risky sexual behavior [24]. A few scholars have adduced the gesture as a subtle way of promoting promiscuity, which has serious detrimental health effects [43,44,45,46]. Thus, logically, the integration of religious morally accurate information into the free disbursement of condom programme could be necessary. 

Most studies excluded examination on the thoughts of the boys/men on pregnancy or fatherhood before engaging in intercourse. The review shows that studies on paternal roles in child upbringing, average cost of child’s schooling, and paternal livelihood are not common. This could imply that researchers are shying away from investigating the opinion of boys/men on paternal coping with unwanted fatherhood, fathers’ attitudes towards a daughter’s unwanted pregnancy, child-bride, child-betrothal, daughter battering (by her husband), and so on. The findings from these types of studies would have represented the exposure of men to the importance of girls and women care, benevolent treatment and support for their rights, not only in terms of sexual and reproductive rights, but also economic and social rights, especially equality and equity in employment or government positions. 

While the popular assertion has been ‘train a girl-child and you train a whole nation.’, literature has not exhaustively dealt with the fact that an untrained boy-child could destroy a whole nation (if not the whole world). In its literary meaning, every stakeholder that could spend a fortune to canvass for girls/women (the nation’s procreators) should not fail to coach a boy-child not to pull such a built-nation down. It could be a colossal investment waste on the protection of sexual and reproductive health rights if the boys and men are not in the forefront. Boys and men should be continuously exposed to counselling and sexuality education including relevant curriculum that dwells on family demography with lessons/modules on the average cost of fathering a child, enlightenment on paternal roles in child upbringing, cost of child’s schooling, unwanted fatherhood, and paternal livelihood. Figure 2 is a schematic diagram that illustrates the possible dividends inherent in the training of boy-children to achieve a safer world for the girls/women. Where the boy-child is exposed to sexual education, responsible living, and counselling, such a boy could be a potential supporter of wanted pregnancy, wanted fatherhood, and marital fidelity. When growing up, such a boy (and eventually a man) would pursue intimate partners’ harmonious living and loyalty, and encourage a wife’s or partner’s empowerment, to list but a few. However, an untrained boy-child could be a potential risk for unwanted pregnancy, unwanted fatherhood, multiple sexual partnerships, women rights violation, gender-based violence e.g., intimate partner violence (IPV) and other adverse sexual behaviour. From the schematic diagram (Figure 2), it is expected that boys or men that protect the rights of girls/women (especially on reproductive and sexual health rights and wellbeing) would not violate such rights, and would, by design, become advocates for such rights. Where boys and men recognise this responsibility (e.g., at homes, in schools, workplace and so on), there would then be appreciable reduction on the ill-outcomes from the untrained boy-child’s actions or behaviours. Therefore, the training of a boy-child could be seen as part of the solution to end sexual violence, unwanted pregnancy, and risky sexual behaviour. The boys and men would now be partners in the adventure to protect girls’ and women’s reproductive health rights and wellbeing [47].

## 4. Conclusions

The study concludes that there is conspicuous silence on boys and men in most programmes related to reproductive health rights. The overriding information suggests that untrained boys and men could constitute a barrier to the enforcement of girls/women rights and could remain a major obstacle in the realisation of sustainable health and wellbeing for girls/women. The authors consider that properly enlightened boys and men could become protectors of girls/women rights (both reproductive and sexual health rights). Men’s and boys’ enlightenment could translate to the realisation of wanted fatherhood, wanted pregnancy, marital fidelity, intimate-partner-softness, and, of course, wife or partner empowerment. The authors suggest that the various stakeholders in girls and women’s wellbeing and nation building, such as teachers/counsellors (in sexuality education), social workers, and the Ministry of Child Education, including community and religious leaders, should extend their concerns to the training of boy-children. There should be counselling not only on the protection of rights, but on the need for responsible sexual behaviours. Where possible, lessons on the cost of fathering a child, enlightenment on paternal roles in child upbringing, the cost of child’s schooling, unwanted fatherhood, and paternal livelihood may be integrated into the school academic activities. Relevant policy to emphasise the minimum age of marriage for boys and men, or the minimum age to start fathering, could be conspicuously highlighted. While the programmes or initiatives that offer access to sexual materials (e.g., free condom distribution) could be sustained, religious information on morals could be integrated into such programmes. In addition, researchers are also encouraged to interrogate more about the boys’/men’s perspectives on paternal coping with unwanted fatherhood, father’s attitudes towards a daughter’s unwanted pregnancy, child-brides, child-betrothal, wife battering, and so on. The outcomes of these studies could help the boys’ and men’s understanding of the possible consequences of the intention to engage in sexual intercourse with a girl-child. Learning experiences from studies that buttressed girls and women coping with unwanted pregnancy could instil caution in the boys before engaging in sexual acts that could lead to unwanted pregnancy.

## Figures and Tables

**Figure 1 ijerph-19-16313-f001:**
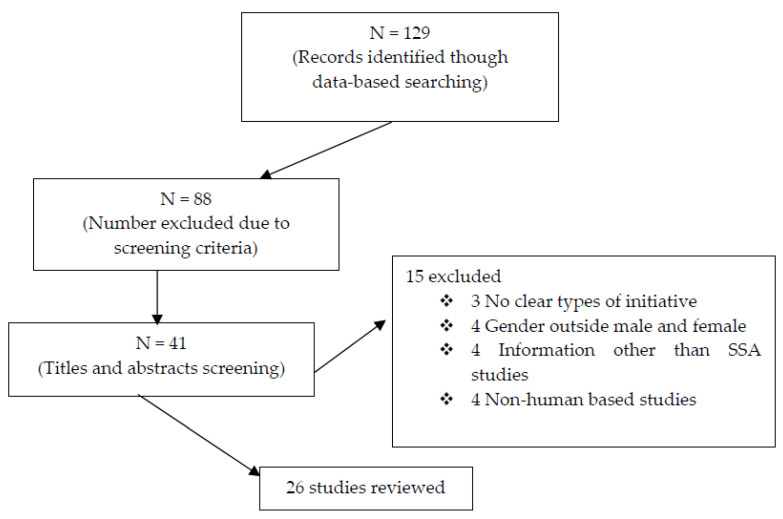
Study selection flow chart.

**Figure 2 ijerph-19-16313-f002:**
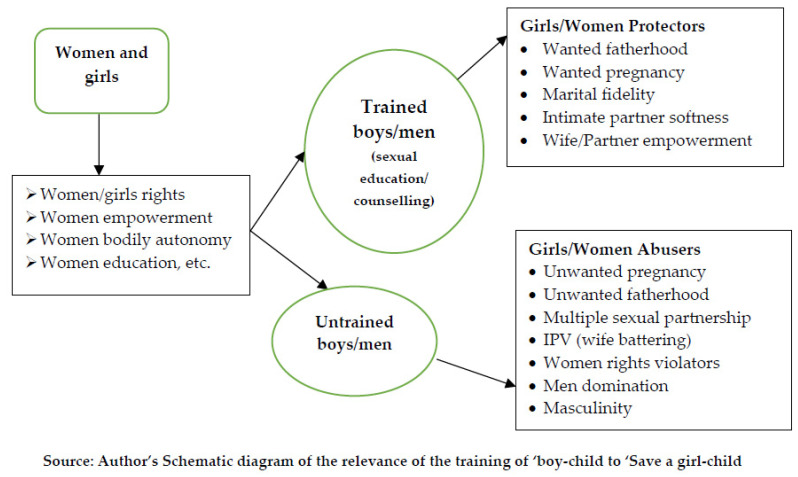
Suggested schematic diagram for ‘Save a girl-child, train a boy-child’.

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
