# Peer review of "To Save a Girl-Child, You Must Train a Boy-Child: A Note on Situational Irony"

_ijerph, 2022, doi:10.3390/ijerph192316313_

Round 1
Reviewer 1 Report
This is a critically important paper as the topic is of great important to SSA and indeed the world. I feel like it could be a much longer paper as it would contribute to the field in many important ways
Some of my comments:
Perhaps a bit simpler language in the paper would be useful and the limit of jargon would be useful. I see how irony is your central theme - both as a political, social and cultural metaphor.
Clarity on how does a nation outlaw child-boy fathering - do you see that through the justice system - incarceration? It needs more clarity. p 3
Your comments on the literature suggesting that promiscuity increases perhaps with condom education could use a bit more in depth analysis of what the literature suggests (is it effective at all?)- p4
You make very important critical points on the premise of the paper about the need to educate they boy-child and the male (who is the perpetrator but absent from almost all education programs). However, it is interesting that you dont make policy recommendations (as promised in your abstract) in a concrete manner. This paper would benefit from a strong conclusion that includes pointed recommendations. The conclusion is weak and does not take the findings to their ultimate goal of concrete policy recommendations based on your analysis of the literature. Who is your audience? If Govt or policy makers are such an audience, then spend some time to giving succinct, clear and hard hitting policy recommendations based on your analysis and the literature. The other audience is scholars who you say have shied away from doing research on boys and men related to reproductive rights. Here too you can make recommendations. how do men
Your schematic (developed by you) gives a somewhat simplistic view of what training accomplishes - if you could complicate that a bit more and make the relevance more far-reaching, e.g. the idea of "protectors" (fathers, sons, brothers, uncles, men generally etc.) still gives men the upper hand in the family system - would training try to create more equality and less gender hierarchy in SSA? Would that be a goal? the idea of "benevolent treatment" also seems to create a ruler/ruled relationship (p. 4). The schematic as a policy recommendation could be fleshed out in greater depth.
The question of polyandry as it relates to South Africa's new law needs your analysis.
Author Response
Response to Reviewers Comments on Manuscript (IJERPH 1916463) entitled “To Save a Girl-Child, You Must Train a Boy-Child: A Note on Situational Irony”
Reviewer No 1
General
We will like to appreciate the reviewers for recognising that the paper focuses on critically topic of great importance to SSA, and indeed the world. We have therefore exhaustively addressed every remark from the reviewers. We hereby present our responses following Query/Observation vs response pattern.
Query: Perhaps a bit simpler language in the paper would be useful and the limit of jargon would be useful. I see how irony is your central theme - both as a political, social and cultural metaphor.
Response: Yes, the use of irony in the paper is very explicit. We are attempting to project .. An irony as the difference between what is being practiced and what is supposed to be the practice. We suspected irony in what is supposed to be equal (or balanced) advocacy but a seemingly one-sided advocacy.
Yes, it speaks to both political, social and cultural contexts of initiatives or intervention that emphasised (or over-emphasised) on one gender and silent on the other.
The word we used are common terminologies in research parlance and we tried to be modest as much as possible.
Query: Clarity on how does a nation outlaw child-boy fathering - do you see that through the justice system - incarceration? It needs more clarity. p 3
Response: The paper suggestions are explicit: enlightenment, encouragement, sexuality education, and conspicuous initiatives that will focus also on the boy-child. For example, we attempted to interrogate why child-bride and adolescent pregnancy are frown at (or seemingly ‘outlawed’), in the same society with passivity (or permissiveness) on child-boy fatherhood? This is for stakeholders to ponder on.
We did not suggest incarceration for child-boy fathering. However, where the anyone (any reader) feels the solution should be ‘incarceration’, then, attempt can be made to write that and publish it. Then the discussion continues….. That’s the secret behind this write up — to spur further discussions or arguments.
Again, the idea of the study is not to provide all the answers to this problem but to further stimulate discussion.
Query: Your comments on the literature suggesting that promiscuity increases perhaps with condom education could use a bit more in depth analysis of what the literature suggests (is it effective at all?)- p4
Response: The paper is not to query the effectiveness of the school-based condom distribution, per se. We only highlighted that there is an element of ‘irony’ in the programme. Specifically, we highlighted that…considering the curiosity inherent in adolescents and young boys, the authors feared (or argued) that exposure of (innocent) school boys and girls to sexual materials (e.g. access to free condom) could spur the attempt to use it. The fear/argument which seems very logical.
We have therefore added that …. scientifically, medically including religious morally accurate information on condom use could be indispensably necessary (to be integrated) within the free-condom disbursement programme.
Again, the we did not specifically interrogate/assess the effectiveness of condom programme but projecting further what could be the implication(s) in terms of continuous inquisitiveness trial of condoms by young people in a world where we want to stop adolescent pregnancy, abortion, etc. It is vivid example of an irony.
Query: You make very important critical points on the premise of the paper about the need to educate they boy-child and the male (who is the perpetrator but absent from almost all education programs). However, it is interesting that you don’t make policy recommendations (as promised in your abstract) in a concrete manner. This paper would benefit from a strong conclusion that includes pointed recommendations.
Query: The conclusion is weak and does not take the findings to their ultimate goal of concrete policy recommendations based on your analysis of the literature. Who is your audience? If Govt or policy makers are such an audience, then spend some time to giving succinct, clear and hard hitting policy recommendations based on your analysis and the literature. The other audience is scholars who you say have shied away from doing research on boys and men related to reproductive rights. Here too you can make recommendations.
Response: We have included policy recommendations in the conclusion section. Specifically, the following have been added:
The authors suggest that the various stakeholders in child wellbeing and nation building such as sexuality education teachers/counsellors, social workers, ministry of child education, including community and religious leaders are to extend their concerns to the training of child-boy. There should be counselling and academic curriculum that dwell on family demogrpahy with lessons/modules on the average cost of fathering a child, enlightenment on paternal roles in child upbringing, cost of child’s schooling, unwanted fatherhood and paternal livelihood. Relevant policy to emphasis the minimum age at marriage for boys and men, or the minimum age to start fathering should be conspicuously highlighted. While the programmes or initiative that offer access to sexual materials (e.g. free condom distribution) could be sustained, the scientifically, medically including religious and morally accurate information on the condom-use could be indispensably necessary (to be integrated) within the programme of free-condom disbursement. In addition, researchers are also encouraged to interrogate more on the boys’/men’s perspectives on paternal coping with unwanted fatherhood, father’s attitude towards daughter’s unwanted pregnancy, child-bride, child-betrothal, daughter battering (by her husband). The outcomes of these research could help in boys and men understanding all possible outcomes of intention to engage in sexual intercourse with a girl-child. Experience to learn (for example) from studies on paternal coping with unwanted fatherhood could instil in the boys the sense of adequate preparation and readiness before engaging in sexual act that could lead to pregnancy.
Query: Your schematic (developed by you) gives a somewhat simplistic view of what training accomplishes - if you could complicate that a bit more and make the relevance more far-reaching.
Response: We cannot claim that our schematic model is exhaustive, No sir. However, we know from experience, that a schematic diagram that will communicate effectively should be simple enough for comprehension by majority of the readers (if not all). We presented our schematic model to narrow the conceived idea and our argument which other researchers can utilise in study broader set of ideas within the context of child-boy training on the issues raised in the paper.
The issues raised in this paper are very sensitive, any attempt to complicate it by a complex scheme or diagram would render the mission of the paper useless. We plea that this is sustained.
Query: The idea of "benevolent treatment" also seems to create a ruler/ruled relationship (p. 4).
Response: No, it cannot. Benevolence treatment is a concept of doing good things for others without expecting anything in return. It is act of treating other very well. It builds trust and reduces quarrels. It is employed here to advocate for selfless protection of girls/women’s rights.
Query: The idea of "protectors" (fathers, sons, brothers, uncles, men generally etc.) still gives men the upper hand in the family system - would training try to create more equality and less gender hierarchy in SSA? Would that be a goal?
Response: The reviewer should note what we wrote is not just protector but….. ‘protectors of girls’/women’s rights’ which we believe, the enlightenment would engender. We envisage such enlightenment to translate into realisation of wanted-fatherhood, wanted-pregnancy, marital fidelity, intimate-partner-softness (and not violence), and of course, wife or partner empowerment.
It is expected that anyone protecting the rights of others (e.g. sexual rights) would not violate such rights and would by designed become an advocate for such rights. Where this is realised at homes, schools, etc, then, there will be appreciable reduction on the ill-outcomes from the untrained boy-child’s actions or behaviours.
Query: The schematic as a policy recommendation could be fleshed out in greater depth.
Response: Done.
Query: The question of polyandry as it relates to South Africa's new law needs your analysis.
Response: Our work specifically focused on what the training of a boy-child could possibly achieve. We know that the paper if published will raise a number of issues for discussion among policy makers, academia and other researchers. However, it is expected that the enlightenment that the boy-child (will receive after this paper) could help them to make informed decision in the future as to whether to choose polygamy or participate in polyandry.
Reviewer 2 Report
Just a few comments
This is a well written paper and extremely significant in today's world.
A few comments:
- calling the boy/ men "culprit" may be a bit too harsh
Also in the conclusions - you need to qualify " The overriding information suggests that men could constitute a barrier" - should read "untrained men"
Your title says child - but the article includes child, adolescent and adult men.
Author Response
Response to Reviewers Comment on Manuscript (IJERPH 1916463) entitled “To Save a Girl-Child, You Must Train a Boy-Child: A Note on Situational Irony”
Reviewer No 2
General
We will like to appreciate the reviewers for recognising that the paper extremely significant in today's world. We have therefore exhaustively addressed every remark from the reviewers. We hereby present our responses following Query/Observation vs response pattern.
Query: - calling the boy/ men "culprit" may be a bit too harsh
Response: Notwithstanding that other reviewer has regarded the term as appropriate, we deemed it fit to use the word: suspected culprit or violators.
Query: Also in the conclusions - you need to qualify " The overriding information suggests that men could constitute a barrier" - should read "untrained men"
Response: Correction effected.
Query: Your title says child - but the article includes child, adolescent and adult men.
Response: Adolescent was mentioned only 5 times in this study: three of the mentions are part of the titles of the references listed, while the 2 mentions are part of results from results from other studies. The focus is on the exposure of boy-child to training (sexual and reproductive counselling, enlightenment on the sexual and reproductive health rights and especially the rights of girls and boys.
Reviewer 3 Report
Very interesting paper that presents a relevant proposal for tackling the problem of reproductive sexuality in sub-Saharan Africa. The proposal should be made more solid by some improvements in the description of the methods and in the language (to facilitate the acceptance of what was also highlighted by others who were still linked to archaic-paternalistic cultural schemes). In particular: the meaning of the exclusion criterion n ° 2 is not clear: how does the position with respect to "reproductive halth and sexual rights" make the data unusable? If the data is unusable, criterion 5 also includes 2! In figure 1 the connection between N = 129 and N = 41 is missing Please enter the definition of the ICPD acronym in the abstract (now on page 2) On page 2 Planned Parenthod is inserted between state and UN organizations; I suggest to delete it as redundant (some might think of a bias in applying criterion 2) The use of the term culprit (which in most cases I think is appropriate) leads to think of an individual responsibility, does not facilitate the understanding of the concept that such behavior can be prevented, try to find some alternative. "other sexual rites" on page 3: referring only to females or both sexes The discussion on the religious aspects only mentions the Pentecostals, who represent a minority of Christianity, it would be interesting and useful in function of the programming of the educational intervention on males, to know what percentage of the population follows religions that prescribe monogamy versus religions that allow polygamy ( for example, are there data on unwanted pregnancies among Christians versus Muslims?).
Author Response
Response to Reviewers Comment on Manuscript (IJERPH 1916463) entitled “To Save a Girl-Child, You Must Train a Boy-Child: A Note on Situational Irony”
Reviewer No 3
General: We appreciate the Reviewer for considering this paper as very interesting paper that is relevant for tackling the problem of reproductive sexuality in sub-Saharan Africa.
We have employed Grammarly tool and other English expert for thorough editing of the manuscript. We also use MS-Word spellcheck.
Query: The proposal should be made more solid by some improvements in the description of the methods and in the language (to facilitate the acceptance of what was also highlighted by others who were still linked to archaic-paternalistic cultural schemes).
Response: We have added a few lines to the methods section to make more detailed.
Query: In particular: the meaning of the exclusion criterion n ° 2 is not clear: how does the position with respect to "reproductive health and sexual rights" make the data unusable? If the data is unusable, criterion 5 also includes 2!
Response: Thanks a lot. Amended to read: studies that are not gendered-based or studies that are not on reproductive and sexual health or rights are excluded.
Query: In figure 1, the connection between N = 129 and N = 41 is missing
Response: Done
Query: Please enter the definition of the ICPD acronym in the abstract (now on page 2)
Response: Added
Query: On page 2 Planned Parenthood is inserted between state and UN organizations; I suggest to delete it as redundant (some might think of a bias in applying criterion 2)
Response: Removed.
Query: The use of the term culprit (which in most cases I think is appropriate) leads to think of an individual responsibility, does not facilitate the understanding of the concept that such behavior can be prevented, try to find some alternative.
Response: We have used the word: suspected culprit or violators.
Query: "other sexual rites" on page 3: referring only to females or both sexes.
Response: One of the examples we have in mind (forced marriages) has been included
Query: The discussion on the religious aspects only mentions the Pentecostals, who represent a minority of Christianity, it would be interesting and useful in function of the programming of the educational intervention on males, to know what percentage of the population follows religions that prescribe monogamy versus religions that allow polygamy (for example, are there data on unwanted pregnancies among Christians versus Muslims?).
Response: We have added a line data on that religion also has been shown to have a significant effect on unintended pregnancy. Specifically, a report has indicated that the risk of untimed pregnancy among women that practice traditional religion is relatively 56% less compared to the Christians and that the likelihood of experiencing unwanted pregnancy among Muslim is relatively 29% lower compared to the rates among the Christian women (Ojuok et al., 2022).
Reviewer 4 Report
The manuscript presents an adequate review study and is an important topic within the area of ​​health and relevant to the geographical area under study. However, authors are required to adjust minor grammatical errors in the document. Also, the author regulations of the journal, indicate that the names of the reference journals must be abbreviated and in the manuscript they are complete.
Author Response
Response to Reviewers Comment on Manuscript (IJERPH 1916463) entitled “To Save a Girl-Child, You Must Train a Boy-Child: A Note on Situational Irony”
Reviewer No 4
General: We appreciate the Reviewer for considering that the manuscript presented an adequate review study and important topic within the area of ​​health that is relevant to the geographical area under study.
Response: Grammatical errors have been corrected. Specifically, we have employed Grammarly tool and other English expert for thorough editing of the manuscript. We also use MS-Word spellcheck.
Query: Reference style
Response: References and in-text citations have been formatted to conform to the journal requirements.
Round 2
Reviewer 1 Report
Please do a complete spell and grammar check, I still see spelling and format errors. Otherwise the paper reads well.